# Evaluation of Patients Who Developed Pneumothorax Due to COVID-19

**DOI:** 10.3390/diagnostics12092140

**Published:** 2022-09-02

**Authors:** Gökhan Çoraplı, Veysi Tekin

**Affiliations:** 1Department of Chest Diseases, Medicine Faculty of Adiyaman University, Adiyaman 02100, Turkey; 2Department of Chest Diseases, Batman Training and Research Hospital, Batman 72100, Turkey

**Keywords:** COVID-19, pneumothorax, chest imaging

## Abstract

Background: Although SARS-CoV-2 infection often affects the lungs, pneumothorax is extremely rare. The aim of this study was to analyze the characteristics of patients who were hospitalized due to COVID-19 and who developed pneumothorax (PNX) and to analyze their risk factors. Methods: Patients who developed PNX, among the patients who were followed up in the hospital due to COVID-19 between 1 April 2020 and 1 April 2022, were included in the study. The mean and standard deviation values of the descriptive statistics were obtained from patient data. The entire application was carried out using IBM SPSS 26 (IBM Corp., Armonk, NY, USA). *p* values < 0.05 were considered statistically significant. Results: We observed that advanced age and male gender increase the risk of COVID-19 patientsdeveloping PNX, but smoking, sepsis, and being followed-up with mechanical ventilation do not increase this risk. In addition, we observed that the presence of an additional disease increases the mortality rate. Conclusion: We show that advanced age and male gender increase the risk for PNX, which is a rare complication of COVID-19, and that comorbidity is associated with mortality in these patients.

## 1. Introduction

SARS-CoV-2 emerged in Wuhan, China, in December 2019. As a result of its rapid spread around the world in spite of the various preventive measures that were taken, it was declared a pandemic by the World Health Organization under the name of COVID-19 on 11 March 2020. The COVID-19 pandemic has caused a number of deaths [1,2]. As of 1 April 2022, the number of more than 490 millioncases had been reported around the world, and the number of deaths has exceeded six million (https://www.worldometers.info/coronavirus/, accessed on 23 July 2022) [3]. A COVID-19 diagnosis is made by evaluating swab samples taken from the nasopharyngeal region using nucleic acid tests. While this disease may be asymptomatic, it may also cause various clinical presentationsthat range from mildsymptoms to respiratory failure [4]. The pulmonary findings of COVID-19 are mostly a bilateral peripheral ground-glass appearance and consolidated infiltrative areas. Although it is less common, the disease can also cause clinical conditions such as pleural effusion and pneumothorax [5].

Pneumothorax (PNX) is found in patients with COVID-19 extremely rarely. A history of smoking, structural lung disease, lung parenchymal injury resulting from strong coughing, alveolar membrane damage, orsudden alveolar pressure increase may cause pneumothorax in COVID-19 patients [6,7]. In addition, for intubated patients, mechanical ventilation and high-pressure oxygen administration are other causes of PNX. A secondary diagnosis of PNX in addition to SARS-CoV-2 infection and early treatment initiation are factors that increase the chances of survival. Publications on this subject are limited, and existing studies have been conducted on a small number of case series. Studies have shown that the incidence of PNX in COVID-19 patients is around 1% [6,7,8]. Because of the scarcity of studies on PNX, which is rare in COVID-19 patients, the present study, which uses an extensive case series, makes a unique contribution to the literature.

## 2. Materials and Methods

### 2.1. Patient Selection

This study was designed as a retrospective observational study and was approved by the local ethics committee (approval code: 30.05.2022/310, 30 May 2022). This study includes a retrospective evaluation of COVID-19 patients who were followed up with in a tertiary regional hospital. Patients who developed PNX within a certain date range were included in the study. Therefore, the sample size could not be predicted.Patients who did not develop PNX were excluded from the study. The study was created by taking data from the hospital database. Patients over the age of 15 years old who were hospitalized in the intensive care unit of our hospital between 1 April 2020 and 1 April 2022 due to SARS-CoV-2 infection were included in this retrospective study. All of the included patients had a confirmed SARS-CoV-2 RNA diagnosis determined by a positive real-time polymerase chain reaction (RT-PCR) and by findings congruent with COVID-19 pneumonia according to thoracic computed tomography (CT) imaging. Patients with negative RT-PCR results for SARS-CoV-2 RNA and those who did not undergo a thoracic CT examination were excluded from the study. The criteria for admission to our hospital were set according to the criteria included in the COVID-19 adult patient treatment guidelines developed by the Ministry of Health of the Republic of Turkey [9].

For patients with clinically suspected PNX, a PNX diagnosis was made by thoracic CT and direct chest X-ray. During follow-up, CT was not considered necessary, and direct radiography was used instead. The images were evaluated by two pulmonologists who had 8 years and 9 years of experience, respectively, in evaluating such images (Figure 1 and Figure 2).

### 2.2. Statistical Analysis

In the statistical analysis phase of the study, frequency analyses and descriptive statistics were utilized. For this purpose, the demographic characteristics of the patients were evaluated, and frequency analyses were performed on the patients with PNX. Frequency (*n*) and percentage (%) values of the groups were calculated within the scope of the frequency analysis. Descriptive statistics of the variables expressing age and day of PNX occurrence were determined. Mean and standard deviation values were obtained from the descriptive statistics. All of the statistical analyses were performed using IBM SPSS Statistics 26 (IBM Corp., Armonk, NY, USA). *p* values < 0.05 were considered statistically significant.

## 3. Results

During the time period in which the patients included in the present study were in the hospital, 3986 patients were admitted to our hospital due to COVID-19, and 37 of those patients developed PNX after hospitalization. Thus, the percentage of patients who developed PNX was 0.92%. Of the patients who developed PNX, 23 (62.2%) were male, and 14 (37.8%) were female. The mean age of these patients was 56.1 ± 17.87 years of age; the youngest patient was 16 years old, and the oldest patient was 88 years old. PNX developed within an average of 9.784 ± 9.678 days during the time in which patients were being monitored for COVID-19. While 17 (45.9%) of the patients were intubated, 20 (54.1%) patients developed PNX when they were in a state of non-invasive ventilation. Pneumomediastinum was accompanying in 8 (21.6%) of the patients who developed PNX. Death due to PNX occurred after 1 day at the earliest and after 18 days at the latest after the occurrence of PNX, and patients were discharged after 2 days at the earliest and after 22 days at the latest after the occurrence of PNX. Table 1 shows the distribution of the patients according to smoking, sepsis status, and PNX characteristics(Table 1).

While 14 (37.8%) of the patients did not have any comorbidities, 23 (62.2%) of them had at least one comorbidity, including respiratory disease, cardiac disease, diabetes, renal failure, or malignancy (Table 2).

The effect of the presence of comorbidities on the day of PNX occurrence was evaluated via a simple regression analysis. Accordingly, it was found that the presence of comorbidities had no effect on the day of PNX occurrence (*p* = 0.226). While 22 (59.4%) of our patients who developed PNX died, 15 (41.6%) of them were discharged and recovered. A Fisher test analysis was performed to examine the effect of the comorbidity status on the mortality of patients with PNX. Accordingly, comorbidity was found to be a factor affecting mortality in PNX patients (*p* = 0.003) (Table 3 and Table 4).

## 4. Discussion

In the present study, the incidence of PNX in our patient population was consistent with the literature. The results of the present study, being similar to all previous studies on PNX in COVID-19 patients, confirm that the data in the present study are accurate. In addition, the rate of PNX in cases of Middle East respiratory syndrome disease caused by SARS-CoV-1, which involves similar respiratory complaints and lung findings, was shown to be 1.7%, thus close to the values for SARS-CoV-2 [10]. In the study of Chen et al. (2020), the incidence of PNX among COVID-19 patients was found to be 1%, similar to the present study. The study by Chen et al.(2020) was one of the first studies on this subject [7]. However, it was carried out in the early stages of the pandemic and only included a small number of cases. Larger-scale studies were conducted after COVID-19 became more widespread. When the images of 6574 patients followed for COVID-19 in 16 centers were analyzed retrospectively, PNX was observed in 60 cases. In thatstudy, the rate of PNX was 0.91% [6]. In a study conducted with 131,679 COVID-19 patients in311 hospitals in England, 1283 PNX cases were observed. In that study, the rate of PNX was 0.97% [11].

In some previous studies, it was observed that 82%, 97%, 73%, and 68% of the patients who experienced PNX due to COVID-19 were male [1,5,9,10]. In addition, the mean age of the patients who developed PNX was found to be between 50 and 70 years old in those same studies [1,6,11,12]. Similarly, in the present study, the patients with PNX were predominantly male, at a rate of 62%. The mean age was found to be 56.1 years old in the present study, once again similar to the literature. The reason why PNX is more common among men and at older ages was not generally hypothesized in previous publications. It may be the result of body features, hormonal structure, a stronger cough reflex, or more exposure to occupational and environmental aerosols.

PNX is a rare manifestation of COVID-19. It is very rare for PNX to be bilateral, and the relevant publications in the literature only cover case reports. In the study conducted by Güven et al.(2021), PNX developed bilaterally in 2 of 10 PNX patients [2]. Likewise, in the study of Wadhawan et al.(2022), PNX developed bilaterally in 2 of 10 PNX patients [12]. Due to the small number of patients in those two studies, the rate was as high as 20%. In studies where the number of patients washigher, this rate decreased. In the study of Martinelli et al.(2020), for example, PNX was bilateral in 4 (7%) of 60 PNX patients [6]. The rate in that study was close to the rate found in the present study (8.1%).

Smoking is considered an important predisposing factor for PNX. However, in previous publications, the majority of PNX patients were non-smokers [6,11,12,13]. In the present study, it was found that 56.8% of the patients who developed PNX did not have a history of smoking, and this result was similar to the results of other studies.

In the present study, the majority of PNX patients (64.9%) did not have sepsis. In the study conducted by Shaikh et al.(2021), the mean Sequential Organ Failure Assessment score was found to be low in COVID-19 patients who developed PNX [1]. Our study showed that sepsis rates were low in patients with PNX, similar to the study of Shaikh et al. (2021). That study is one of the few available publications on this subject. Therefore, further research is needed on this subject.

In the study of Belleti et al. (2021), it was shown that mechanical ventilation was a predictive factor for PNX in patients with COVID-19 [14]. In addition, mechanical ventilation was seen as a risk-increasing factor in many studies examining the characteristics of COVID-19 patients who developed PNX [1,2,6,11]. However, unlike other studies, the majority of patients who developed PNX (54.1%) were in a state of non-invasive ventilation.

The present study showed that there was no significancebetween the presence of comorbidities and the day that PNX developed in patients who were hospitalized in the intensive care unit (*p* = 0.226). It was also demonstrated that there was significancebetween PNX-related mortality and the presence of comorbidities (*p* = 0.003). Another unique aspect of the present study is the lack of previous studies in the literature revealing correlations among the presence of comorbidities, the day of PNX development, and mortality.

When PNX develops in hospitalized patients due to COVID-19, the severity of the disease increases, and, as a result, there is an increase in mortality rates. Previous studies on this subject [11,12,15] and the findings of the present study (59.4%) strengthen this theory.

The limitations of the present study were that it was single-centered; not all of the factors that may facilitate PNX were checked; and our patients were not compared to patients who developed spontaneous PNX regardless of their COVID-19 status. Further prospective randomized controlled multi-center studies are needed to fully understand the relationship between COVID-19 and PNX.

## 5. Conclusions

In the present study, it was found that PNX was more common in male and elderly COVID-19 patients, and its relationship with sepsis, smoking history, and mechanical ventilation was weaker than expected.Although there are studies on this subject, the large number of cases and the original evaluations carried out during the analysis of these cases make this study unique.We think that this study, which examines PNX, a very rare side effect of COVID-19, will make an important contribution to the literature.

## Figures and Tables

**Figure 1 diagnostics-12-02140-f001:**
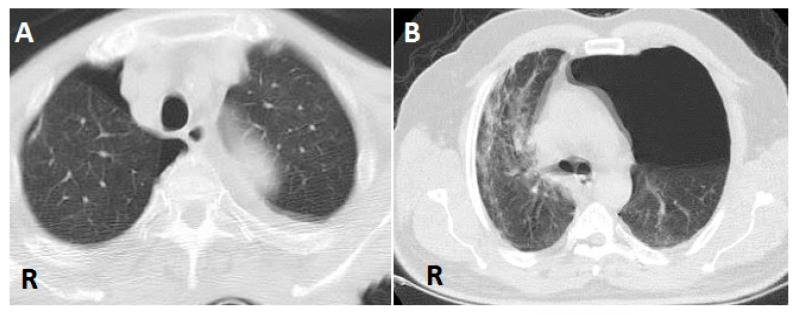
(**A**) Air is seen in the right pleural space due to pneumothorax (PNX) on axial thoracic CT scan. (**B**) Air is seen in the left pleural space due to PNX on axial thoracic CT scan (R: right).

**Figure 2 diagnostics-12-02140-f002:**
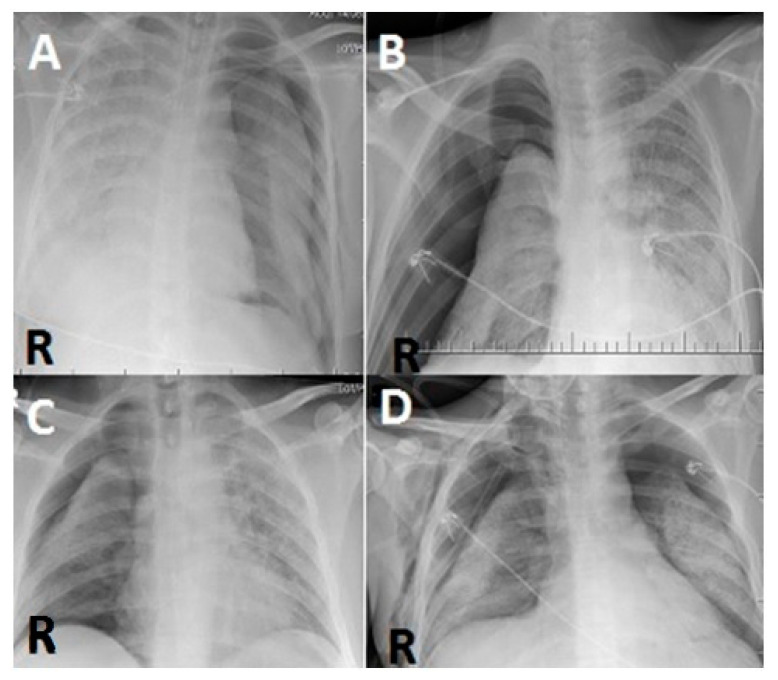
(**A**) In a posterior–anterior lung roentgenogram, the right lung is almost completely radiopaque, while PNX is seen on the left side. (**B**,**C**) In a posterior–anterior lung roentgenogram, the left lung is almost completely radiopaque, while PNX is seen on the right side. (**D**) Bilateral PNX is observed in aposterior–anterior lung roentgenogram (R: right).

**Table 1 diagnostics-12-02140-t001:** Demographic characteristics of the patients and their distribution according to the pneumothorax (PNX) characteristics (*n* =37).

Variable	*n*	%
** *Gender* **		
Male	23	62.2
Female	14	37.8
** *Age, years ** **	56.108 ± 17.877	
** *DaystoPNX occurrence ** **	9.784 ± 9.678	
** *Status during PNX* **		
Intubated	17	45.9
Non-invasive ventilation	20	54.1
** *Ex/discharge status after PNX* **		
Exitus, 1st day	1	2.7
Exitus, 2nd day	7	18.9
Discharge, 2nd day	2	5.4
Exitus, 3rd day	1	2.7
Exitus, 4th day	3	8.1
Discharge, 4th day	1	2.7
Exitus, 5th day	2	5.4
Discharge, 5th day	1	2.7
Exitus, 6th day	1	2.7
Discharge, 7th day	2	5.4
Discharge, 8th day	2	5.4
Exitus, 9th day	1	2.7
Exitus, 10th day	1	2.7
Discharge, 10th day	3	8.1
Exitus, 11th day	1	2.7
Discharge, 13th day	1	2.7
Discharge, 14th day	1	2.7
Exitus, 15th day	2	5.4
Exitus, 18th day	2	5.4
Discharge, 22nd day	2	5.4
** *PNX uni/bilateral* **		
Bilateral	3	8.1
Unilateral	34	91.9
** *Presence of septic shock* **		
No	24	64.9
Yes	13	35.1
** *Smoking* **		
No	21	56.8
Yes	16	43.2

* Mean ± standard deviation.

**Table 2 diagnostics-12-02140-t002:** Distribution of patients according to the presence of comorbidities (*n* =37).

Variable	*n*	%
** *Comorbidity* **		
None	14	37.8
Cancer	7	18.9
Diabetes mellitus (DM)	1	2.7
Epilepsy	1	2.7
Pregnancy	1	2.7
Hypertension (HT)	2	5.4
Chronic obstructive pulmonary disease	1	2.7
Chronic kidney disease	3	8.1
CVD (cardiovascular disease)	7	18.9

**Table 3 diagnostics-12-02140-t003:** Regression analysis results of the effects of comorbidity presence on day of PNX occurrence.

Variable	Group	Beta	SE	t	*p*
Comorbidity	Yes	[Ref]	-	-	-
	No	4.140	3.358	1.233	0.226
Constant		8.217	2.066	3.978	<0.001
R^2^ = 0.042					
F-statistic = 1.520					
*p* value = 0.226					

Beta: coefficient; SE: standard error; R^2^: coefficient of determination; t: time (day).

**Table 4 diagnostics-12-02140-t004:** Association between presence of comorbidity and exitus/discharge status after PNX.

Variable	Presence of Comorbidity	
	Yes	No
Exitus, 1st day	4.35% (1)	0.00% (0)
Exitus, 2nd day	30.43% (7)	0.00% (0)
Discharge, 2nd day	0.00% (0)	14.29% (2)
Exitus, 3rd day	4.35% (1)	0.00% (0)
Exitus, 4th day	0.00% (0)	21.43% (3)
Discharge, 4th day	0.00% (0)	7.14% (1)
Exitus, 5th day	8.70% (2)	0.00% (0)
Discharge, 5th day	4.35% (1)	0.00% (0)
Exitus, 6th day	0.00% (0)	7.14% (1)
Discharge, 7th day	4.35% (1)	7.14% (1)
Discharge, 8th day	8.70% (2)	0.00% (0)
Exitus, 9th day	4.35% (1)	0.00% (0)
Exitus, 10th day	4.35% (1)	0.00% (0)
Discharge, 10th day	4.35% (1)	14.29% (2)
Exitus, 11th day	4.35% (1)	0.00% (0)
Discharge, 13th day	0.00% (0)	7.14% (1)
Discharge, 14th day	4.35% (1)	0.00% (0)
Exitus, 15th day	8.70% (2)	0.00% (0)
Exitus, 18th day	4.35% (1)	7.14% (1)
Discharge, 22nd day	0.00% (0)	14.29% (2)

Note: As a result of the Fisher test, the *p* value was found to be 0.003.

## Data Availability

Data will be made available upon appropriate request.

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
