# Peer review of "Evaluation of Patients Who Developed Pneumothorax Due to COVID-19"

_diagnostics, 2022, doi:10.3390/diagnostics12092140_

Round 1

Reviewer 1 Report

I read this article with great interest. This manuscript is well-written and well-designed. However, substantial changes should be made before acceptance.

- Data about non-invasive ventilation should be included and appropriate correlations should be made.

- Data about  pneumomediastinum are missing and should be included and analyzed as well.

- Did authors estimate sample size? Please specify in data analysis section.

- I found several English errors throughout the paper. Please have a deep language revision.

Author Response

REVIEWER 1 :

 Comments and Suggestions for Authors

General Comment:  I read this article with great interest. This manuscript is well-written and well-designed. However, substantial changes should be made before acceptance.

RESPONSE: Thank you for your thorough review and comments. The manuscript was revised thoroughly taking into account your suggestions and comments. The changes are highlighted in the manuscript.

  1. Data about non-invasive ventilation should be included and appropriate correlations should be made.

RESPONSE: We retrospectively found that 37 of 3986 patients who were hospitalized in our hospital between April 2020 and April 2022 developed PNX while being treated under intensive care conditions. In 17 of 37 patients, PNX developed while in the invasive ventilation (intubated) state. The remaining 20 patients developed PNX while they were in the non-invasive ventilation state. We have described this situation as extubated due to our mistake in the manuscript. This is our fault. The term Extubated has been changed to non-invasive ventilation. Appropriate correlation analyzes were performed.

  1. Data about  pneumomediastinum are missing and should be included and analyzed as well.

RESPONSE: We did not mention it in our study because we thought that pneumomediastinum (n: 8) was not found in a considerable number among the PNX patients included in our study, and we thought that it would not contribute to the article. We also wanted our writing to be focused on PNX. However, if you wish, we are willing to mention this situation in the article.

  1. Did authors estimate sample size? Please specify in data analysis section.

RESPONSE: Since this study was designed as a retrospective observational study, the sample size was not estimated by the authors. This study retrospectively analyzed the COVID-19 data of a territory hospital.

  1. I found several English errors throughout the paper. Please have a deep language revision.

RESPONSE:  In line with your suggestion, the language revision of the journal was made using Language Editing Services. If you think that current revision of the text is not clear for the readers, we are willing to further discuss this aspect and to make more changes.

Reviewer 2 Report

Need revisions before this manuscript can be published

Please refer the attached manuscript for the comments

Author Response

REVIEWER 2 :

 Comments and Suggestions for Authors

General Comment:  Need revisions before this manuscript can be published. Please refer the attached manuscript for the comments.

RESPONSE: Thank you for your thorough review and comments. The manuscript was revised thoroughly taking into account your suggestions and comments. The changes are highlighted in the manuscript. If you thinks that current revision of the text is not clear for the readers, we are willing to further discuss this aspect and to make more changes.

- Title rephrased to reflect study content.

-Added a short intro to the Background section.

- Requested reference changes have been made

- Keywords rearranged

-The typos that were requested to be corrected in the introduction have been corrected. In addition, new important information has been added to the introduction.

- The methodology section has been rewritten in detail.

-Criteria for admission to our hospital were set according to the criteria included in the COVID-19 adult patient treatment guidelines of the Ministry of Health of the Republic of Turkey (Reference 9)

- The typos that were requested to be corrected in the methodology part have been corrected

- Requested changes were made in the Result section and it was made more understandable.

- We have updated table 1 and table 2 based on your suggestions.

- Correlation is misspelled due to translation error in the title of Table 4. Changed to Association.

- In Table 4, we conducted cross-tab analysis. We change the term correlation with association. These are the categorical variables, so we could construct a cross-tab. In this cross-tab, some cells had very low values, due to this fact, we used Fisher test to check the significance. Fisher test can be applied for rxc (r: number of rows, c: number of columns) tables as well, not only 2x2 tables.

-We have updated the figures based on your suggestion.

- The typos that were requested to be corrected in the discussion part have been corrected.

- With the changes made in the discussion section in line with your suggestions, the article has been made more clear.

- Changes were made in the discussion section in line with your requests and suggestions for typos. Incomprehensible expressions have been changed. Language corrections have been made. In addition, structural changes were made to make the discussion part clearer for the readers.

- In line with your suggestion, the language revision of the journal was made using Language Editing Services. If you think that current revision of the text is not clear for the readers, we are willing to further discuss this aspect and to make more changes.

Round 2

Reviewer 1 Report

The revised version is significantly improved.

Please add data about pneumomediastinum and the manuscript in my opinion can be accepted.  

Author Response

RESPONSE: Thank you for your valuable review and comments. The manuscript was revised taking into account your suggestion and comment. Data about pneumomediastinum has been added to the results section. The changes are highlighted in the manuscript.

Reviewer 2 Report

This manuscript is publishable

Author Response

RESPONSE: Thank you for your valuable review and comment.